# Exploring canine's olfactive threshold in artificial urine for medical detection

Michelle Leemans[1*◦], Sara Hoummady[2◦], Emmanuelle Boutin[1], Adeline Giganti[3], Laetitia Maidodou[4,5], Vincent Cuzuel[6], Sabrine Ajili[3], Damien Steyer[4], Caroline Gilbert[7,8], Isabelle Fromantin[3]

**1** Clinical Epidemiology and Ageing Unit, Institut Mondor de Recherche Biomédicale, INSERM, Paris-Est University, Créteil, France, AP-HP, Hopital Henri-Mondor, Clinical Research Unit (URC Mondor), Créteil, France, **2** Institut Polytechnique UniLaSalle, IDEALISS ULR 7519, Université d'Artois, Mont Saint Aignan, France, **3** Wound Care and Research Unit 26, Curie Institute, Rue d'Ulm, Paris, France, **4** Twistaroma, Illkirch Graffenstaden, France, **5** DSA, IPHC UMR7178, Université de Strasbourg, Strasbourg, France; CITHEFOR, EA 3452, Université de Lorraine, Nancy, France, **6** Forensic Institute of the French Gendarmerie, Caserne Lange, 5 Boulevard de l'Hautil, Cedex, Cergy-Pontoise, France, **7** Ecole Nationale Vétérinaire d'Alfort, 7 avenue du Général de Gaulle, Maisons-Alfort, France, **8** Laboratoire Mecadev, UMR 7179, CNRS/MNHN, 1 avenue du Petit Château, Brunoy, France

◦ These authors contributed equally.
* michelleleemans91@gmail.com

## Abstract

Canine olfaction is increasingly studied as a tool for detecting cancer and other diseases. Previous pilot studies have demonstrated that dogs can effectively distinguish positive samples from negative samples in humans with breast cancer, achieving sensitivity rates as high as 100%. However, questions remain about dogs' ability to detect low concentrations of volatile organic compounds in complex medium. While dogs' detection thresholds for isoamyl acetate using a mineral oil substrate have been studied, there are no current studies on their detection limits using more complex substrates like urine, relevant in clinical settings. This pilot study aimed to evaluate the olfactory threshold of dogs using artificial urine with various concentrations of isoamyl acetate.

Two dogs were trained to detect isoamyl acetate, initially using water as the substrate during the training phase, and subsequently using artificial urine during the testing phase, under single and double-blinded conditions. The dogs were trained to indicate the presence of isoamyl acetate solutions by sitting in front of the positive sample and ignoring controls. Training and testing occurred in a controlled environment, maintaining consistency with the same two handlers, a standardized methodology, and positive reinforcement with toy rewards. Based on double-blind performances, results showed a minimum detection threshold of 6.7 x 10-9 Molar (M) for Nougaro (Springer Spaniel) one dog and 2.1 x 10-7M for Prince (Labrador Retriever). The sample age did not affect performance. However, the position of the cone did, with higher failure

**Data availability statement:** The dataset and its corresponding data dictionary are publicly available on Figshare at https://figshare.com/s/d128cd03724e5be874ee.

**Funding:** This study was funded by the Royal Canin Foundation and a sponsorship from Seris Security. The funders had no role in the study design, data collection, analysis, or interpretation of the results.

**Competing interests:** The authors have declared that no competing interests exist.

rates for the first cone compared to the other three. These findings underscore the potential of trained dogs to detect volatile organic compounds at very low concentrations in complex substrates, supporting their use in clinical diagnostics.

## Introduction

In 2022, cancer affected nearly 20 million people worldwide and resulted in about 9.7 million deaths, with lung cancer being the most prevalent, followed by breast, colorectal, and stomach cancers [1]. Although screening programs are in place, they sometimes miss early-stage cancers and often target specific age groups, such as 50–65 years of age, leaving gaps in early detection and management for other populations. Additionally, not everyone has access to these screening programs, particularly in resource-limited settings.

Dogs have demonstrated the ability to detect a range of complex targets, including narcotics, explosives, and various diseases [2]. Recent research has explored the potential of canine olfaction as a complementary cancer detection method [3]. Their potential for identifying cancers—such as breast, cervical, colorectal, ovarian, and stomach—has garnered significant interest due to their high sensitivity and specificity [4]. Canine detection sensitivity ranges from 17% to 100%, with specificity varying based on individual dogs' anatomical and training factors [5]. This capability is linked to their ability to detect volatile organic compounds (VOCs) associated with disease states [6].

VOCs, low molecular compounds that easily evaporate at room temperature [7], show great potential for non-invasive cancer screening using trained dogs. While gas chromatography coupled with mass spectrometry (GC-MS) remains the gold standard for VOC detection, its practical application is limited by high costs, time-intensive processes, and lack of portability [6]. In contrast, using dogs for VOC detection offers a simpler, non-invasive, and potentially low-cost method. VOC canine screening can be performed in various settings, making it accessible even in resource-limited environments. The canine olfactory detection threshold—the minimum concentration at which an odorant can be reliably detected—varies widely [8]. For example, thresholds for amyl acetate range from 40 parts per billion (ppb) to 1.5 parts per trillion (ppt) in mineral oil [8], possibly influenced by factors such as genetic variation, training, methodology [9,10] or even microbiota [11,12]. To put 1 ppt into perspective, it is equivalent to dissolving one grain of sugar in an Olympic-sized swimming pool. One of the major drawbacks of using dogs in large-scale canine detection programs, such as COVID-19, is their limited concentration span, which we have set to 30 minutes per session. However, by optimizing the screening process, many samples can be processed within this time frame, making the method potentially efficient and promising for large-scale use.

The choice of the substrate is crucial in experimental setups, particularly because some biological samples (i.e.,: urine, sweat) are valuable in clinical settings for potential early detection, and this despite their challenges for dogs due to their

complex mixtures of VOCs. While these substrates hold significant clinical potential, and urine being a frequently used substrate for cancer detection, they have not yet been studied for canine detection thresholds. In contrast, mineral oil is frequently used in experimental studies due to its simplicity, despite its lack of relevance for clinical applications.

Within this pilot study, we trained two dogs to investigate the canine detection threshold for a single VOC within a complex mixture. We focus on isoamyl acetate, selected for its consistency and comparability in prior studies of canine olfaction [8,13–15]. Our objectives were to define the artificial urine, including identifying the VOCs to be added and evaluating the stability of the chosen mixture, and to determine the canine detection threshold of isoamyl acetate in this artificial urine mixture, comparing the results with findings from studies using mineral oil as a solvent.

## Materials and methods

### Dogs

This study involved two detection dogs from the KDOG Program: Nougaro, a 7-year-old intact Labrador Retriever weighing 34 kg, and Prince, a 5-year-old intact Springer Spaniel weighing 19 kg. Both dogs had previous experience in medical detection. Both dogs were fed the same commercially available dry diet (4300 Sporting Life Energy, Royal Canin SAS, Aimargues, France).

### Odor sample preparation

**Literature review and selection process.** A comprehensive literature review was undertaken using PubMed and Google Scholar to identify VOCs present in human urine. The objective was to identify the most frequently detected and quantifiable VOCs using gas chromatography-mass spectrometry (GC-MS) techniques. A selection of 12 relevant publications was made for analysis, detailed in S1 Table. The focus of the literature review was on identifying VOCs consistently reported across multiple studies. This approach ensured the representation of diverse chemical families and included the most cited VOCs from the selected 12 publications. S2 Table provides comprehensive information on the compounds identified in the selected publications, including their total number of references among the 12 publications, concentrations found in human urine, corresponding references, and a brief description as well as their probable origin. Following the identification of key VOCs, a careful selection of 12 compounds was made to create an artificial urine mixture. This selection process aimed to replicate the composition of human urine while ensuring a balanced representation of the most frequently detected VOCs. List of VOCs selected to compose artificial urine: 3-hexanone; 2-heptanone; 4-heptanone; 2-pentanone; 2-butanone; p-cresol; dimethyldisulfide; furan; octanal; pyrrole; p-cymene; trimethylamine.

**Preparation of artificial urine.** To achieve accurate concentrations and mimic physiological levels of VOCs found in human urine, a protocol adapted from Sarigul et al [16]. was followed. The creation of an artificial urine matrix closely resembling human urine involved two main steps. Firstly, a basic urine matrix was prepared by dissolving multiple compounds in distilled water at 37°C. The pH was adjusted to 5–6, and the sterility of the artificial urine matrix was ensured using sterile filtration units (0.2 μm, Fisherbrand polyethersulfone). Furthermore, pH levels of the basic urine matrix were monitored for 6 weeks to ensure adherence to physiological ranges commonly observed in human urine. In the second step, the 12 additional identified VOCs were added to the basic urine matrix either just before or up to a maximum of 1 week before sample usage (see S3 Table for composition and concentration). For compounds whose concentration in urine has not been determined (p-cymene; octanal; dimethyl disulfide; pyrrole), a default concentration of 5 nmol/mmol creatinine was assigned. Following the addition of these l2 VOCs, stability tests were conducted to evaluate the robustness of the matrix over time, ensuring consistency and reliability in subsequent analytical procedures. An analysis was conducted using Stir Bar Sorptive Extraction (SBSE) on 1 cm length magnetic stir bars (Twisters, Gerstel) coated with polydimethylsiloxane (1 mm film thickness) and GC-ToF-MS (GC 7890B, Agilent coupled with Pegasus BT

ToF-MS, LECO) to confirm the stability of the mixture. The peak area ratios, which represents the intensity ratio between a compound and an internal standard, was used to assess the stability of compounds over time. The comparison of peak area ratios between the initial analysis at T0 and the analysis at t + 7 days (expressed as a percentage) showed no variation, indicating consistent signal intensity. This suggests that the compounds remained stable, with no detectable degradation, during 7 days of storage at 4°C (S1 Fig).

**Sample preparation.** The dogs were trained to detect solutions of Isoamyl acetate (or Acetic acid 3-methylbutyl ester or Isopentyl acetate, CAS 123-92-2; ≥ 99.7% Sigma Aldrich, 79857–5ml, 6.73mol/L) diluted in water or artificial urine at different concentrations. Isoamyl acetate was chosen based on previous studies testing olfactory detection thresholds in humans [17], rodents [18,19], and dogs [13,14,20] but also because of its high volatility, its distinct banana-like odor, and safety to handle. The isoamyl acetate stock solution is diluted in half-log or quarter-log in distilled water ensuring consistency in the preparation of the target odor (serial dilution). This cascade dilution involved multiple steps, with each step consisting of 5 µL of the stock solution being mixed with the appropriate volume of distilled water to achieve 10 mL of the desired concentration. The number of dilutions (ranging from 1–6) depended on the desired final concentration, with each step using the solution from the previous dilution. The first concentration presented to the dogs is $6.73 \times 10^{-4}$ molar (M), during the training phase. All serial dilution steps were made in clean 15 centrifuge tubes, PET, conical bottom w/plug seal cap (Product# CLS430055, Merck). Samples were prepared in advance for a full week of work with the dogs.

**Analysis of isoamyl acetate stability in aqueous medium.** To investigate the stability of isoamyl acetate in water, solutions were prepared at three concentrations ($6.7 \times 10^{-6}$ M, $6.7 \times 10^{-7}$ M, and $6.7 \times 10^{-8}$ M). Each concentration was replicated three times at each time point for analysis.

The samples were stored at + 4°C and analyzed at three intervals: immediately after preparation (t0), after 9 days, and after 17 days. Static headspace analyses were performed with a GC-MS Q2010 Ultra purchased from Shimadzu (Kyoto, Japan). A RTX-5MS column (50 m × 0.18 mm, 0.17 µm) (Restek, France) was used to conduct the chromatographic separation. The initial temperature was 35 °C, held for 4 min, first raised to 140 °C at 5 °C/min and then raised to 280 °C at 12.5 °C/min and held for 10 min. A mass spectrometer was used with the electronic ionization source (70 eV) heated at 200 °C. The acquisition was made with scan and SIM modes. The scan range was 30–270 m/z. The SIM window was from 12 to 12.50 minutes and the ions 43, 55, 70, and 87 were monitored. Data were acquired with GC Real Time Analysis and processed with GC PostRun Analysis 4.53 (Shimadzu software).

**Analysis of isoamyl acetate stability in artificial urine.** To assess the stability of isoamyl acetate in artificial urine, solutions were prepared at three different concentrations: $6.729 \times 10^{-6}$ M, $6.729 \times 10^{-7}$ M, and $6.729 \times 10^{-8}$ M. Three technical duplicates were performed for each concentration at each time point. The tubes were stored at + 4°C. Samples were analysed at three time points: immediately after preparation (t0), after 9 days, and after 19 days. Solid Phase MicroExtraction (SPME) combined with GC-MS (Agilent 7890B GC,5977 MSD System) on a DB-624 semi-polar column (30 m x 0.32 mm x 1.4 µm) (Agilent) was used for analysis. A DVB/CAR/PDMS SPME fiber was employed for the sample extraction. Raw data were processed using MassHunter® Quantitative Analysis software. p-Cresol, present in the matrix at $0.2854 \times 10^{-6}$ M and stable in the model urine, served as the internal standard.

## Training and testing procedure

Training and tests were conducted between May 2023 and July 2023 in Champvoisy, Marne, France, using two combined Algeco modular units, which are portable, prefabricated buildings used for various purposes. This setup created a controlled environment that allowed for the separation of the dog and handler from the operator (the person who knew the location of the positive sample during single-blind session). A fixed-position camera captured all sessions, and data were simultaneously uploaded to an online server for analysis.

The training was conducted by two experienced handlers. The dogs were trained using positive reinforcement methods and rewarded with toys. They were exercised five days a week, receiving twice-daily off-leash walks in groups with

alongside other dogs. Dogs participated in up to two sessions per day, one conducted in the morning before lunch and another in the afternoon. The dogs had training and testing in a temperature- (average 20°C) and humidity-controlled (humidity between 40%-60%) experimental room over 5 months, working with the same handler.

During both training and testing phases, four metal cones were mounted on a support at a height of 72 cm for the Labrador Retriever and 39 cm for the Springer Spanish, enabling the dogs to insert their noses into the cones. The cones containing the detection samples are designed to enhance the diffusion of VOCs by featuring a wide and low opening (Fig 1A). Each session consisted of multiple lines, with each line comprising four different cones (Fig 1B).

In the initial training phase, cones contained either water with isoamyl acetate or water alone. Later in the experiment, in both the training and testing phases, the cones were filled with either artificial urine alone or artificial urine mixed with isoamyl acetate.

To enhance VOC diffusion, 10 mL of isoamyl acetate solution of a given concentration was placed in low, wide-mouth glass containers. Three control cones contained 10 mL of artificial urine in identical, thoroughly cleaned glass containers, with each container cleaned with water after each line. Clean water was used to remove isoamyl acetate residues between lines, as its odorless nature prevents interference, unlike alcohol/ethanol, which could affect the dogs' reactions. At the end of the day alcohol was used to clean the cones.

Dogs were initially trained using water as a substrate with various concentrations of isoamyl acetate, and the same protocol was applied to both training with water and testing with artificial urine. The target odor location was pseudorandom

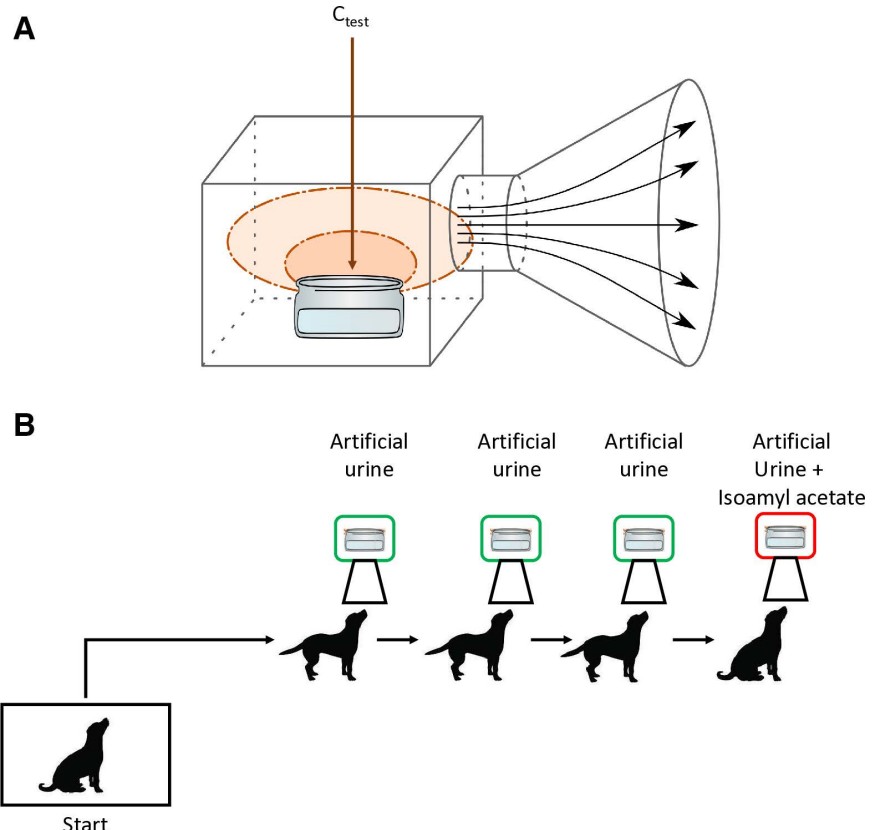

**Fig 1. A. A costum-made metal cone with a wide opening and space for a wide-mouth vial containing the sample. B. Example of the experimental setup for training and testing environment, with the positive sample positioned on the 4th cone.**

such that the target odorant was not in the same location for more than three trials in a row. The experimental design was divided into two different sessions (Fig 2):

1. **Training session (Fig 2 A):** Each training session began with an evaluation of the dog's interest. This involved three initial tests at a concentration of isoamyl acetate higher than the test concentration (C > 1/2 log (Ctest)). Subsequently, the dogs were tested with at least four test lines using the target concentration in a single-blind setup. Each line consisted of four cones, and between each line, the samples were replaced and the cones were cleaned with water.

2. **Test session (Fig 2 B):** The test session consists of three lines of interest, two or three lines of simple blind tests, then four lines of double-blind tests. In the preliminary single-blinded phase, we required dogs to correctly complete two trials before advancing to the double-blinded phase. If a dog succeeded in only one out of the first two trials, a third trial was conducted to provide an additional opportunity for qualification. During the test sessions, if the dog achieved a success rate of 75% (correctly identifying the positive sample in at least 75% of the tests), the concentration was decreased by ½ log or ¼ log for subsequent training and test sessions.

If a dog failed to show adequate interest—indicated by failure in 2 or 3 of the initial interest tests—a one-hour break was implemented before a new session. If failure persisted after the break, a corrective session with a higher concentration could be conducted later the same day if signs of lack of interest were observed.

The testing protocol included both single-blind and double-blind conditions:

• **Single-Blind Tests:** The handler was unaware of the location of the positive sample, the position of the positive sample is known only to the operator in the adjacent building unit. After passing the single-blind tests, dogs moved to double-blind testing. In practice, there is a window between the two units that allows the operator to communicate with the handler. This setup enables the handler to observe the operator's cues and reward the dog correctly when it makes the right decision, or withhold rewards if the dog makes an incorrect decision.

• **Double-Blind Tests:** Neither the handler nor the operator knew the location of the positive sample. These sessions included three lines for assessment of interest, followed by two or three single-blind lines and then 4 double-blind lines.

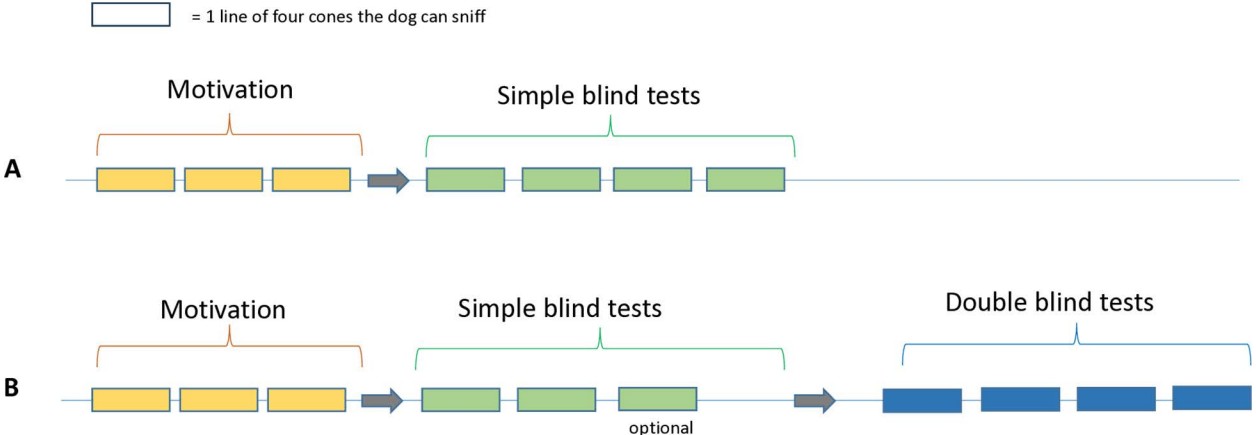

**Fig 2. Experimental design of training and testing sessions.** A. Training session: Dogs were trained during 1–4 days with three lines of motivation of followed by a simple blinded lines with the tested concentration. When the dog successfully passes the level of concentration during testing (success in more than 75% of the lines), he was tested in a double-blind situation (B). B. Testing session: The testing session consists of three lines of motivation, two or three lines of simple blind tests, then four lines of double-blind tests. In the preliminary single-blinded phase, we required dogs to correctly complete two trials before advancing to the double-blinded phase. If a dog succeeded in only one out of the first two trials, a third trial was conducted to provide an additional opportunity for qualification.

Samples for both single- and double-blind sessions were prepared in the same manner, with the only difference being the level of anonymization. Samples were prepared in advance at the laboratory, and tubes for double-blind tests were anonymized. During double-blind sessions, the dog was always rewarded.

It is crucial to recognize that biases can arise if someone involved knows the results, as dogs are highly sensitive to subtle cues. To address this, meticulous precautions were taken to minimize unintentional signaling and maintain the integrity of the testing conditions.

The dogs were trained, with a clicker, to sit in front of the cone containing the isoamyl acetate sample and wait for a reward (a toy). In case of an incorrect indication, the dog was not rewarded. In double-blind situations, dogs were rewarded for each indication since the handler was unaware of the positive sample's location.

The detection threshold for each dog was determined when the success rate dropped below 75% (Fig 3). In case of failure, another double-blind session (4 lines) is tested after a corrective session with a higher concentration. The maximum amount of trials was a maximum of 4 with each trial (=line) consisting of 4 cones.

**Weather data analysis**

The potential impact of weather conditions on the results was analyzed using the meteorological data from two nearby weather stations provided by Météo France (https://meteo.data.gouv.fr/datasets/donnees-climatologiques-de-base-horaires/). The stations, Chambrercy (station number 51111001) and Igny-Comblizy (station number 51298001) are located within 20 kilometers of the test site in Champvoisy, France (department 51).

Due to the unavailability of weather data for the Chambrercy station for the year 2023, the data from the Igny-Comblizy station were used. The data were segmented into hourly intervals, with these one-hour periods selected as the most appropriate for analysis. The analysis specifically examined variables of temperature and precipitation.

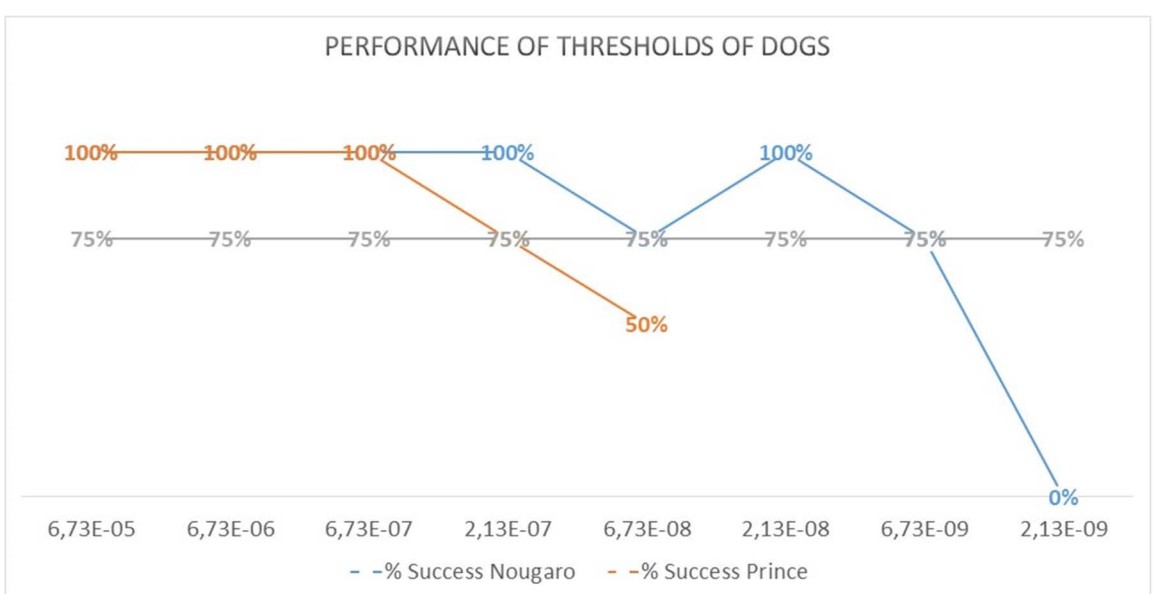

**Fig 3. Performance of two dogs relative to concentration levels.** On the y-axis are the performances of the dogs in %, by concentration. The concentrations are indicated on the x-axis The success threshold is set at 75%.

## Canine behaviour

A list of behaviors assessed by video.was developed to ethically assess some stress behaviours of dogs during tests, based on a previous publication [21] (S4 Table). Behaviors were coded from video recordings using the software Boris [22] (version 8.27 – Università di Torino). Additionally, a simpler observation of the dogs' behaviours was coded by two independent experimenters, following the criteria outlined in Table 1. We attempted to observe which paws were lifted; however, the camera position did not allow for sufficient precision. We only have accurate information for the first two paws. The objective of these observations was to assess the dog's emotional state.

## Statistical analysis

Qualitative data are expressed as numbers and percentages [n (%)] and quantitative data as median (interquartile range [IQR]). Data were compared using Pearson's chi-square test or Fisher exact test. To determine the optimal session duration that maximizes the success rate, we utilized a Receiver Operating Characteristic (ROC) analysis. We then plotted the ROC curve by calculating the true positive rate (sensitivity) and the false positive rate (1 - specificity) for different duration thresholds. In this context, sensitivity refers to the ability of the trained dogs to correctly identify the presence of the target VOC in the complex mixture (true positive rate), while specificity refers to the ability of the dogs to correctly identify the absence of the target VOC (true negative rate). These measures help assess the accuracy of the dogs' detection performance. The Area Under the Curve (AUC) was computed to assess the model's discriminative capability. The optimal threshold was determined using the Youden Index, defined as $J = Sensitivity + Specificity - 1$, to maximize the sensitivity and specificity values. All tests were two-tailed and $P < 0.05$ were considered statistically significant. Analyses were performed with STATA v17.0 (StataCorp, College Station, TX, USA).

## Ethical approval

The two handlers strictly followed ethical guidelines under the supervision of a veterinarian from the French National Veterinary School of Alfort (CG). The dogs used in the study were privately owned and regularly monitored by veterinarians to ensure their health and well-being. The environment met all welfare standards, and the study was approved by the ethical committee of Royal Canin, the study's sponsor. Data and video recordings were used with the handlers' consent, and no biological samples from either dogs or humans were collected during the procedure.

## Results

### Artificial urine preparation

An artificial urine matrix was successfully developed to replicate physiological levels of VOCs in human urine. To minimize bias, pseudorandomized target locations were employed for sample presentation, and all samples were pre-prepared in

**Table 1. Coding scheme for behaviours associated with marking.**

| Dogs' behaviour | Definition |
|---|---|
| Correct search | The dog did everything in the way it was expected to do. |
| Indecision | The dog hesitates or performs a partial mark (lowers its hindquarters but quickly raises them again) on one or more cones. |
| No selection | The dog sniffed several cones but does not signalling any cone on its own. |
| Do not search | The dog does not seem to sniff the cones or repeatedly looks at the handler during the run. |
| Sweep | The dog inspected cone x+1 and then returned to inspect the previous cone x. This behaviour was noted for further analysis. |
| Look | The dog looked at the handler before indicating. |

the laboratory to ensure consistency throughout the training sessions. Over a six-week monitoring period, pH levels were confirmed to be maintained within physiological ranges (data not shown).

### Analysis of isoamyl acetate stability in squeous medium

In water, isoamyl acetate signals decreased on average by approximately 84% after 9 days with complete degradation observed after 17 days at 4°C for the highest concentration (see S2 Fig).

### Analysis of isoamyl acetate stability in artificial urine

In artificial urine, isoamyl acetate signals decreased on average by approximately 25% after 9 days and 37% after 19 days of storage at 4°C, with p-cresol serving as a stable internal standard throughout the analysis (see S3 Fig).

### Description of tests

A total of 76 sessions were conducted by the 2 dogs, with a mean duration of 24 minutes (range: 19–28 minutes). In total (including interest, simple-blinded tests, and double-blinded tests), 597 lines were performed, with 407/579 (68.2%) lines being positively marked and 52/65 (80%) during the double-blinded tests. Regarding the dogs' behaviour, 19 lines (3.2%) were not marked by the dogs, and the dogs appeared indecisive in 49 (8.2%) the lines. One dog exhibited slightly higher performance, succeeding in 6 out of 7 sessions during the double-blind test, compared to 7 out of 9 sessions for the second dog. The mean age of the samples was 6.5 days (range: 3.5–7 days) for the double-blind test, compared to 6 days (range: 5–7 days) for the overall sessions (Table 2).

### Dogs' detection threshold of amyl acetate in artificial urine

The lowest olfactory threshold for isoamyl acetate achieved was $6.7 \times 10^{-9}$ M in artificial urine for Nougaro (Labrador Retriever) and $2.1 \times 10^{-7}$ M for Prince (Springer Spaniel) (only the tests in double-blinded conditions were used for this analysis). Water was used solely for training the dogs to detect isoamyl acetate odor.

### Influence of sample age on performance

In the double-blind data, the dogs' behavior ($p = 0.549$) nor performance ($p = 0.1$) was influenced by the age of the sample (Table 3)

### Performance in double-blinded tests

Dogs marked the first cone in the line more frequently ($p = 0.049$, Table 4), despite the position of the positive cone being correctly randomized during double-blind tests ($p = 0.72$, data not shown). None of the behaviors observed using the ethogram-like table (S4 Table) were linked to performance in double-blind tests (data not shown).

### Behavior observations in single-blind tests

In single-blind tests, dogs looked at the handler less frequently after marking when they successfully identified the correct sample, underscoring the importance of focusing on the dogs' behavior during tests ($p = 0.008$, data not shown). Sweeping behavior did not affect the performance of the dogs ($p = 1$, data not shown).

### Impact of weather conditions

The potential impact of weather conditions on the results was assessed using meteorological data from the Igny-Comblizy weather station, as data from the Chambrercy station were unavailable for 2023. Hourly temperature and precipitation data were analyzed, revealing no significant seasonal variations or external temperature fluctuations that affected the

**Table 2. General overview of the results, including the total number of trials and double-blinded conditions. Results are presented as absolute values, with square brackets indicating the range (minimum to maximum) and percentages shown in parentheses.**

| | |
|---|---|
| **TOTAL LINES** | |
| Number of dogs | 2 |
| Number of sessions | 76 |
| Duration of session (min) (n = 76) | 24 [19–28] |
| Age of the sample (days) (n = 76) | 6 [5–7] |
| Number of cones | 2388 |
| Number of lines | 597 |
| Number of positives lines marked (n = 597) | 407 (68.2) |
| Behaviour of the dog per line (n = 597) | |
| No comment | 511 (85.6) |
| Indecision | 49 (8.2) |
| No selection | 19 (3.2) |
| Does not search | 18 (3.0) |
| **DOUBLE-BLIND LINES** | |
| Number of sessions | 16 |
| Number of successful sessions (n = 16) | 13 (81.3) |
| Number of successful sessions NOUGARO (n = 9) | 7 (77.8) |
| Number of successful sessions PRINCE (n = 7) | 6 (85.7) |
| Age of the sample (days) (n = 16) | 6.5 [3.5–7] |
| Number of cones | 260 |
| Number of lines | 65 |
| Number of positive lines marked (n = 65) | 52 (80.0) |
| Behaviour of the dog per line (n = 65) | |
| No specific observation | 63 (97.0) |
| Indecision | 1 (1.5) |
| No selection | 1 (1.5) |
| Does not search | 0 (0) |

**Table 3. Dogs' behaviour and performance linked to sample age during double-blinded tests. Percentages are shown in parenthesis.**

| | <5 days | [5–7[days | ≥7 days | p |
|---|---|---|---|---|
| Behaviour of the dog<br>No specific behaviour linked to decision | 20/20 (100) | 16/17 (94.1) | 27/28 (96.4) | 0.549 |
| Indecision | 0 (0) | 0 (0) | 1/28 (3.6) | |
| No selection | 0 (0) | 1/17 (5.9) | 0 (0) | |
| Performance | | | | |
| Success | 3/4 (75) | 2/4 (50) | 8/8 (100) | 0.100 |

**Table 4. Description of position of negative marked samples in urine, in double-blinded settings. Percentages are shown in parenthesis.**

| | Position 1 n = 65 | Position 2 n = 65 | Position 3 n = 65 | Position 4 n = 64 | p |
|---|---|---|---|---|---|
| Cones with negative samples that were marked by the dogs | 6 (9.2) | 4 (6.2) | 2 (3.1) | 0 (0) | 0.049 |

dogs' detection abilities (data not shown). Since the dogs are from outdoors and often play outside between sessions, we measured all factors, including weather, to account for potential influences on their behavior and ability to detect VOCs. Additionally, we wanted to verify that the Algeco modular units were suitable for this type of setup.

## Discussion

This study evaluates dogs' ability to detect isoamyl acetate in a urine-like mixture, showing they can reliably identify it in the complex medium at very low concentrations ($6.7 \times 10^{-9}$ molar – 6.7 parts per billions). Detection becomes more challenging as concentrations decrease. A previous study on isoamyl acetate in mineral oil found a detection threshold between 40 ppb to 1.5 ppt, with significant inter-dog variability [8]. Our findings align with this, but our study set a higher success threshold at 75%, compared to the 40% threshold used by Concha et al. [8]. Dogs' detection threshold for isoamyl acetate in both artificial urine and mineral oil is at the ppb level, highlighting their exceptional sensitivity and potential in high-sensitivity medical tasks.

Initial instability in water led us to test it in urine, where it proved more stable. Aging of amyl acetate did not affect detection ability, but chemical stability must be rigorously analysed in future VOC studies to avoid unreliable results.

To ensure reliable results and minimize bias, both single-blind and double-blind protocols were used. The behaviour differences in the two dogs during single-blind and double-blind tests highlights the importance of these setups. Previous studies [23,24] have shown that handler knowledge affects team performance and dogs' behaviour. In single-blind setups, the operator, aware of the target odor location, was isolated to prevent interaction with the dog, minimizing unintended cues. However, the handlers' knowledge in single-blind protocols could still influence the dog's behavior. Double-blind protocols eliminated potential biases from handlers [24], reinforcing the study's integrity. Future studies should implement the training method recommended by Edwards et al. (2017) [25], where the operator with sample status information is positioned outside the room and uses a numerical screen to communicate with the trainer. This approach would help avoid reinforcing false indications for disease-negative samples, as observed in this study, and strengthen the validity of the findings.

This study highlights performance differences between the two dogs, reflecting olfactory capacities influenced by factors such as genetics [9,26], trainability [27], age [28], breed [29], medications [30], and microbiome composition[11]. These differences emphasize the need for standardized methodologies in canine medical detection and a larger number of dogs to better understand the factors that influence outcomes.

Throughout the study, the dogs exhibited minimal stress, though camera positioning may have affected behavioural assessments. The two dogs involved in this protocol showed no behavioural issues, but more anxious dogs may take longer to solve tasks and exhibit stress-related behaviours [31], highlighting the importance of assessing a dog's temperament before selecting it for detection tasks [32].

Developing a reliable canine working protocol is essential for establishing detection thresholds, as many dogs are excluded during training, especially when detecting low-concentration biomarkers for diseases like cancer. Using artificial mixtures in selection protocols could help screen dogs without depleting valuable biological samples. Due to our small size (only two dogs), we could not validate this protocol. Future studies should include a larger, more diverse dog cohort, test across various diseases and matrices, and include medically relevant odors to refine and validate the protocol for real-world medical detection applications.

We conducted the study in a mobile environment with two interconnected modular units, offering flexibility in relocating the unit as needed and particularly beneficial in resource-constrained countries. We maintained strict control over temperature and humidity levels within the unit. Our findings indicated that seasonal variations and external temperature fluctuations did not impact the dogs' detection abilities within this mobile setup, aligning with results from previous studies [28,33].

The spatial organization of odour cones impacted detection accuracy, with dogs consistently marking samples within the first cone. While target odor positions were randomized, complete randomization can reinforce positional bias if the dog favours a particular location. Personalized randomization could help adapt to each dog's tendencies [34]. A circular

setup (carousel) [8] has been proposed to enhance detection reliability by allowing dogs to revisit samples. Optimizing cone organization could improve canine detection accuracy in practical applications.

In conclusion, our study confirms the remarkable ability of dogs to detect amyl acetate in complex mixtures within a mobile and controlled environment. These findings align with prior studies on the exceptional sensitivy of canine olfaction, even in challenging conditions. Notably, this is the first pilot study to explore a dog's ability to detect a single VOC in a complex medium like artificial urine, advancing research into canine medical detection. Despite variability between the two dogs tested, our findings showcase the unique capabilities of canine olfaction and its potential to complement non-invasive diagnostic tools in medical research.

## Supporting information

**S1 Table. Publications for selection of key volatile organic compounds in human urine.**
(XLSX)

**S2 Table. Identified VOCs in a literature search of 12 relevant publications with their total number of references, chemical family, found concentration, and short description.**
(XLSX)

**S3 Table. Composition of artificial urine.**
(XLSX)

**S4 Table. List of behaviors assessed by video.**
(XLSX)

**S1 Fig. Bar charts showing the relative intensities of GC-MS peak areas of VOCs in the model urine matrix at the initial time ("t0") and after 7 days of storage at 4°C ("t0 + 7 days"). The peak areas at the initial time ("t0") are expressed as 100% reference values (blue bars). The "Ratio t0 + 7 days/t0" (red bars) indicates the percentage ratio of values obtained after 7 days of storage to the initial values. Error bars represent the average standard deviation (n = 3).**
(TIF)

**S2 Fig. Stability of amyl-acetate in water. The boxplot displays the interquartile range, with the central line representing the median. The whiskers extend to the range of the data (1.5 IQR from the quartiles), and the diamond-shaped dots indicate outliers. The missing part of the boxplot corresponds to 0 values, which are not visible due to the log scale used in the plot. The Y-axis represents the area under the curve refers to the integrated signal corresponding to a specific compound or ion as it elutes through the chromatographic column. This area is used to quantify the amount of a particular substance in the sample. The X-axis represents the theoretical concentration of Isoamyl Acetate in water.**
(TIF)

**S3 Fig. Stability of amyl-acetate in artificial urine. The boxplot displays the interquartile range, with the central line representing the median. The whiskers extend to the range of the data (1.5 IQR from the quartiles), and the diamond-shaped dots indicate outliers. The missing part of the boxplot corresponds to 0 values, which are not visible due to the log scale used in the plot. The Y-axis represents the relative response in GC-MS, which is the intensity of the analyte signal relative to the total ion current or to an internal standard, allowing for comparison across different concentrations and conditions. The X-axis represents the theoretical concentration of Isoamyl Acetate in urine.**
(TIF)

## Acknowledgments

The authors wish to thank the Royal Canin Foundation, Handi'Chiens, Didier Valentin, Florian Conchaudon, Philippe Mont, Camille Alahdef, Elyes Hadjsaid and Hind Baba Aissa for their support and assistance with this project.

## Author contributions

**Conceptualization:** Caroline Gilbert, Isabelle Fromantin.

**Data curation:** Michelle Leemans, Adeline Giganti, Laetitia Maidodou, Vincent Cuzuel, Sabrine Ajili.

**Formal analysis:** Michelle Leemans, Sara Hoummady, Emmanuelle Boutin.

**Funding acquisition:** Isabelle Fromantin.

**Investigation:** Michelle Leemans, Laetitia Maidodou, Vincent Cuzuel, Sabrine Ajili, Isabelle Fromantin.

**Methodology:** Michelle Leemans, Adeline Giganti, Laetitia Maidodou, Damien Steyer, Isabelle Fromantin.

**Project administration:** Adeline Giganti.

**Supervision:** Adeline Giganti, Caroline Gilbert.

**Validation:** Sara Hoummady.

**Visualization:** Michelle Leemans, Sara Hoummady.

**Writing – original draft:** Michelle Leemans, Sara Hoummady, Adeline Giganti.

**Writing – review & editing:** Emmanuelle Boutin, Laetitia Maidodou, Vincent Cuzuel, Sabrine Ajili, Isabelle Fromantin.

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
