## [Decision Letter · Decision Letter 0]

13 Dec 2024

PONE-D-24-48244Exploring canine’s olfactive threshold in artificial urine for medical detectionPLOS ONE

Dear Dr. Leemans,

Thank you for submitting your manuscript to PLOS ONE. After careful consideration, we feel that it has merit but does not fully meet PLOS ONE’s publication criteria as it currently stands. Therefore, we invite you to submit a revised version of the manuscript that addresses the points raised during the review process. The methods should be improved according to the reviewers' comments.

We look forward to receiving your revised manuscript.

Kind regards,

Etsuro Ito, Ph.D.

Academic Editor

PLOS ONE

Journal Requirements:

2. In the online submission form, you indicated that [Data supporting the findings of this study are available from the corresponding author upon reasonable request.].

Reviewers' comments:

Reviewer's Responses to Questions

**Comments to the Author**

1. Is the manuscript technically sound, and do the data support the conclusions?

Reviewer #1: Partly

Reviewer #2: Partly

2. Has the statistical analysis been performed appropriately and rigorously? 

Reviewer #1: Yes

Reviewer #2: Yes

3. Have the authors made all data underlying the findings in their manuscript fully available?

Reviewer #1: Yes

Reviewer #2: No

4. Is the manuscript presented in an intelligible fashion and written in standard English?

Reviewer #1: Yes

Reviewer #2: Yes

5. Review Comments to the Author

Reviewer #1: We need more data papers on working dogs and this very small - really a pilot study - can be a part of those data. The paper needs some work.

Lines 125 – 126 – Please take out ‘being the first’ – you have 2 dogs with widely different performances – you an say nothing about generality. Just say “We trained 2 dogs….et cetera…”. Really, this should more appropriately be called a pilot study.

Lines 138-139 This is an inadequate description of their training – be specific.

Line 142 I don’t know what the KDOG centre is and most readers will not either

Line 239 “Motivation” is likely the wrong word here – you were assessing a dog’s interest (and many detection dog guidelines make this clear) the compound. There are a number of criteria used to evaluate motivation and you meet none of them. Please change wording throughout.

Line 271, 274 et cetera – ‘Frustration’ is another one of those words that is overly used and seldom defined. You did not reward them each time to decrease ‘frustration’ – however you define this. You did so because when you cannot know if the compound is there (eg, accelerants) you use a continuous reward schedule. You used this for 2 reasons: because you could not know in that phase (and this is why training where someone tells you in a double blind that you were correct is so critical – something that was apparently not done here and I cannot fathom why not) AND you were rewarding the dogs for working – for making a choice. You may want to read and cite some of the work of Adee Schoon on methodology for detection dogs.

Line 291 – These authors did not create this ethogram to assess stress. Instead, this was an exhaustive ethogram of behaviors exhibited under a set of 3 request conditions. Working dogs dropped many of these out compared to pet dogs. Please correct the wording.

Line 300 – Why don’t you define sensitivity and specificity first. Many people get it wrong.

Line 332 – Again – remove motivation and use ‘interest’.

Lines 349-350 – It’s not statistically significant. STOP HERE. You have 2 dogs. It’s not biologically meaningful to them or any subject who would hope to benefit from their efforts.

Lines 383+ - I am a big fan of a good literature review – but you had 2 dogs. You can talk about what other authors have opined but there is no relevance of any of this here to you with 2 dogs.

Lines 385 + - Your observed differences highlight the need for an adequate sample size.

Lines 404+ - Here’s where commentary by Schoon and Jezierski could really help you.

Line 415 – Circular set up ref??

Lines 435 + - This a very lengthy discussion for what is essentially a pilot study – you do not need most of this opinion about protocols et cetera. Condense all of this to the essentials.

Also - please check that formatting and language are correct throughout for an English language journal. For example - quotes should be on "top".

Reviewer #2: General comments:

This study examined two dogs' olfactory detection threshold for isoamyl acetate, which have been studied earlier. However, the authors created a more practical/lifelike situation by creating a stable artificial urine solution (which is a great achievement) and tested to what extent these two detection dogs were able to identify the isoamyl acetate in it.

However, there is one major problem with the study. There is no difference between the single-blind and double-blind conditions. The authors stated that in single-blind tests: “The handler was unaware of the location of the positive sample, the position of the positive sample is known only to the operator in the adjacent building unit.” Then they state in the discussion: “In a single-blind situation, the handler, being blind, is the one who can provide clues that may subtly affect the dog's behavior. Double-blind protocols, while more complex to execute, provided additional assurance by eliminating potential biases from human handlers, thereby reinforcing the study's methodological integrity.”

In single-blind protocols only the operator knows the location of the target, but the operator is present in the test location (See Script 3 of Experiment 2 (page 3) in DeChant et al. (2020)! In this case the operator or the dog may be able to read subtle cues from the operator (not the handler who has no idea about the target location!) about the location of the target scent.

In double-blind protocols, neither the handler, nor the operator knows the location of the target scent, so the dog or the handler would not be able to read subtle cues about the location of the target from the operator. So by eliminating the presence of the operator next to the line-up during the single-blind protocol, the authors effectively made the situation double-blind.

Based on this the paper needs a major rewrite, in a way that the single-blind tests are either left out or merged with the double-blind test data and then a reanalysis is also needed.

Apart from the above, the introduction and discussion are relatively well written, but the methods and results need clarifications. The tables and graphs also need further clarifications.

Other specific comments:

L78: olfactory threshold OF dogs

L81-83: This should be rephrased as it sounds that the dogs were separately trained to indicate each different concentration of isoamyl acetate. However, I presume the dogs were trained to indicate the presence of isoamyl acetate.

L83: “consistent handlers” – consistent in what way?

L84-85: “During double-blinded tests, handlers were unaware of the target’s position.” – this is common knowledge, so it can be deleted

L86: “Be consistent in the use of comma or dots for decimals. Use the one required by the journal.”; Do the dogs not have an ID number or names? Why are they just called one dog and the other?

L109-111: This is all true, however, it is important to bear in mind that a dog can work a couple of hours before it needs to rest. The amount of dogs that would be necessary for a cancer screening programme can be enormous. Covid detection dogs had the same hype, and this was one of the reasons, why they were not utilized in large scale, whether it be hospitals or public places.

L126: What is the relevance of isoamyl acetate from a medical point of view? Why did not the authors use some molecule that is medically relevant for cancer detection? What is the reason why many other studies used isoamyl acetate for olfactory detection threshold?

L131-132: This last sentence would fit more in the conclusion and not in the aim of the study.

L137: “were intact” – is already mentioned in the previous sentence

L139-140: “They were exercised five days a week, receiving twice-daily off-leash walks in groups with other dogs and training sessions.” – This sentence is confusing, should be rephrased. Clarify how many training sessions and walks were done daily.

L141: delete comma before “4300”

L142: duplicate bracket

L143: what was the temperature and humidity setting in the experimental room?

L143-144: “working with the same handler.” – It is mentioned above that the training was conducted by two experimenters, so please clarify.

L162: Dot is missing from the end of the sentence. Why is the Furan spelled with a capital F, while the other compounds are not?

L166: The year should be deleted.

L181: What is peak area ratio?

L189: Keep it consistent, and use isoamyl acetate throughout the text to avoid confusion, if isoamyl acetate and isopentyl acetate are the same.

L193-195: If we take 5 microliter of stock solution and mix it with distilled water to achieve 10 ml of the desired solution, then that is only one concentration. It would be good to at least mention how many dilution steps were done.

L197: there should be a multiplication sign here instead of a dot

L199-200: at this point the reader does not even know what is the experimental setup, so this sentence is a but out of context here

L219: why was the last time point different in the artificial urine compared to water?

L226: delete “in 2023”, it is mentioned after that twice

L227: what is an Algeco modular unit?

L236: there is no Figure 1C

L243-244: How did the authors ensure that the clean water (relatively odorless) removed every residue of the isoamyl acetate solution (target scent) from the container?

L250-251: If there were four cones and at least six trials, how did the trainers ensured that the dogs will not favor one cone for another? E.g. in case of six trials the target scent could have been in cone no 1 and 2 twice, and in cone no 3 and 4 only once, which could have increased for preference for cones no 1 and 2 in the line-up.

L254-256: how many trials were allowed at each concentration level to reach pass the criterion?

L267: Dogs could only achieve 75% success in single blind tests (2 or 3 trials), if they were correct in all trials. Right?

L271: dog WAS always rewarded

L279-288: What was the reason for the weather data analysis? Based on L230, I thought the tests were done indoors in a temperature and humidity controlled room.

L292: citation is missing for BORIS (Friard and Gamba, 2016)

L305: there is an extra “s” at the end of the sentence

L311: Is there an approval number?

L315-321: The preparation of the artificial urine is already described in the methods.

L323-325: Any numbers or statistics about the significant decrease would be appreciated here.

L326-329: Any statistical info would be appreciated here.

L332-335: It would be easier for the reader if the results would be reported consistently in the text (either all % in brackets, or all lines in brackets).

L340-342: I don't understand this sentence. According to the methods, there were 3 dilutions used: 6.729 × 10⁻⁶ M, 6.729 × 10⁻⁷ M, and 6.729 × 10⁻⁸ M. So how could the yielded threshold be anything else for the two dogs?

L344-345: at least a p value or some numbers would be appreciated here

L347: “Dogs marked the first cone in the line more frequently” – I don't really see this from Table 4, but what I see is that the dogs were allowed to go through the line-up only once, as they were quite likely to signal at the last cone.

L348-349: This table is about sweeping, and the p value does not match in the text and the table.

L352: “data not show” – Table 3 seems to contain such information?

L353-358: The impact of weather conditions is already mentioned in the methods. I don't even know what is the point of this part, if the tests were done indoors.

L370-371: why is that study less stringent, than the current study? the same principles were used to find the detection threshold by lowering the concentration of the target odor

L411-412: “with a selection of the bowl directly in front of the experimenter” – I would remove this statement, as the experimenter was not standing next to the bowl in this study; that part is not in accordance with the results. The operator was isolated in a separate part of the unit.

L451-453: I would also add "conducting tests with medically more relevant odors".

L499: Table 1:

Wouldn't it be better to call the “No comment” the “correct search”, when the dog did everything in the way it was expected to do?

No selection: I would suggest using “dog sniffing” instead of “dog tested”, also “marking” is when a dog urinates on something therefore I would change that to “signalling”. “this comment does not apply” – This is a bit confusing remark here. The non-selection does not apply?

Sweep: So this was not allowed? The dogs had to signal on the first occasion in the correct cone?

L500: Table2: Be a bit more specific in the table caption, at least mention that the results of all the lines and the double-blind lines, and also stat what is in the brackets.

L501: Table 3: Based on the table caption, I would say this table is redundant, and the p values could be added to the text. However, if the authors wanted to show how the age of the samples affected the dogs' behavior and performance, then they should state it in the table caption. Further, state what is in the brackets.

L508: Table 5: What is the point of this table? It could be easily described in one sentence in the text.

L520: Figure 1A: Based on this picture, the reader might think that the target scent was always located in the last in the line-up. More description is needed here. Further, it is confusing to see isopentyl acetate when the authors mention isoamyl acetate almost everywhere in the text. If isoamyl acetate and isopentyl acetate are the same (or can be used as synonyms) then the authors should stick to one of them to avoid confusion.

L522: Figure 2: What makes the third single-blind test optional?

L523: Figure 3: The text is tiny on this figure, it is barley readable.

L546: missing journal name and issue number

L551, L554, L562, L601, L605, L610: missing page numbers or article number

Supplementary Table 2: What is BDD? Is the total number of references relate to the Supplementary Table 1? Is the urine concentration range created from the lowest and highest concentration reported, or the lowest and highest reported average? In several rows in the compound name does not match the name mentioned in the description. In several descriptions it seems the beginning of the description is missing, or it begins with a half sentence, or the compound name is missing.

Supplementary Table 3: In Supplementary Table 2 dots were used as decimals, while here it is a comma. Please make it consistent.

Supplementary Table 4:

Panting: OPEN mouths

Is the conductor the operator?

Avoiding physical contact: who is the someone who would touch the dog during testing?

Lowering head: lowering to what extent, how much?

Barking, whipper, backwards ears, lifts the leg while sniffing: event instead of even; lifts which leg? Why is it important?

Play with the handler: what is real play mean? Is there unreal play interaction too? what did they do for playing?

Supplementary Table 5: There is no need for this table. The information is so little that it could be just written in the text.

Supplementary Figure 1: Where is the figure caption? What do we see on the figure? What is peak area ratio, what is TO, what do the whiskers show?

Supplementary Figure 2: Where is the figure caption? What do the boxplots show (lines, whiskers, diamonds shaped dots)? Where is the rest of the green boxplot, or is the bottom of it is 0? What is the measure of the area, squared meter, squared centimeter etc.? Theoretical concentration of what (x axis description)?

Supplementary Figure 3: Same problem, as with Supplementary Figure 2

6. PLOS authors have the option to publish the peer review history of their article (what does this mean? ). If published, this will include your full peer review and any attached files.

**Do you want your identity to be public for this peer review?** For information about this choice, including consent withdrawal, please see our Privacy Policy .

Reviewer #1: No

Reviewer #2: No

---

## [Author Response · Author response to Decision Letter 1]

19 Feb 2025

Dear,

Thank you for the constructive comments and suggestions. We have revised the manuscript in accordance with the reviewers' recommendations. Below, you will find our detailed responses to each comment. Additionally, we have attached an updated version of the manuscript with track changes highlighting the modifications made in response to each reviewer’s feedback.

Kind regards,

Michelle Leemans

---

## [Decision Letter · Decision Letter 1]

5 Mar 2025

Exploring canine’s olfactive threshold in artificial urine for medical detection

PONE-D-24-48244R1

Dear Dr. Leemans,

We’re pleased to inform you that your manuscript has been judged scientifically suitable for publication and will be formally accepted for publication once it meets all outstanding technical requirements.

Kind regards,

Etsuro Ito, Ph.D.

Academic Editor

PLOS ONE

Reviewers' comments:

Reviewer's Responses to Questions

**Comments to the Author**

1. If the authors have adequately addressed your comments raised in a previous round of review and you feel that this manuscript is now acceptable for publication, you may indicate that here to bypass the “Comments to the Author” section, enter your conflict of interest statement in the “Confidential to Editor” section, and submit your "Accept" recommendation.

Reviewer #2: All comments have been addressed

2. Is the manuscript technically sound, and do the data support the conclusions?

Reviewer #2: Yes

3. Has the statistical analysis been performed appropriately and rigorously? 

Reviewer #2: Yes

4. Have the authors made all data underlying the findings in their manuscript fully available?

Reviewer #2: Yes

5. Is the manuscript presented in an intelligible fashion and written in standard English?

Reviewer #2: Yes

6. Review Comments to the Author

Reviewer #2: General comments:

The authors addressed all my comments, so I am happy with the manuscript.

Few specific comments:

L88: delete “one dog”

L265: replace “Spanish” with “Spaniel” – Springer SPANIEL

L346-347: It is still unclear what is the connection between dogs' emotional state and lifting their paws?

L541: A more up to date study from that group about effects of breed (instead of Polgár et al. 2016) is this: Salamon et al. 2025 (https://doi.org/10.1038/s41598-025-87136-y), which also shows that the personality of the dog also affects olfaction.

L572: on the exceptional SENSITIVITY

L637: CUSTOM-made metal cone

L722: Jenkins reference does not have page/article number. Front Vet Sci, 5, ... (2018).

L751: DeChant reference does not have page/article number. Front Vet Sci, 7, ... (2020).

L781: Lazarowski reference does not have page/article number. Front Vet Sci, 7, ... (2020).

7. PLOS authors have the option to publish the peer review history of their article (what does this mean? ). If published, this will include your full peer review and any attached files.

**Do you want your identity to be public for this peer review?** For information about this choice, including consent withdrawal, please see our Privacy Policy .

Reviewer #2: No

---

## [Editor Report · Acceptance letter]

PONE-D-24-48244R1

PLOS ONE

Dear Dr. Leemans,

I'm pleased to inform you that your manuscript has been deemed suitable for publication in PLOS ONE. Congratulations! Your manuscript is now being handed over to our production team.

Kind regards,

on behalf of

Prof. Etsuro Ito

Academic Editor

PLOS ONE